# Neoadjuvant Approaches to Non-Melanoma Skin Cancer

**DOI:** 10.3390/cancers15235494

**Published:** 2023-11-21

**Authors:** David C. Wilde, Mica E. Glaun, Michael K. Wong, Neil D. Gross

**Affiliations:** 1Bobby R. Alford Department of Otolaryngology–Head and Neck Surgery, Baylor College of Medicine, 1977 Butler Blvd. Suite E5.200, Houston, TX 77030, USA; 2Department of Head and Neck Surgery, The University of Texas MD Anderson Cancer Center, 1515 Holcombe Blvd., Houston, TX 77030, USA; 3Department of Melanoma Medical Oncology, The University of Texas MD Anderson Cancer Center, 1515 Holcombe Blvd., Houston, TX 77030, USA; mkwong@mdanderson.org

**Keywords:** neoadjuvant therapy, basal cell carcinoma, cutaneous squamous cell carcinoma, Merkel cell carcinoma, non-melanomatous skin cancer

## Abstract

**Simple Summary:**

Non-melanomatous skin cancer (NMSC) is the most incident malignancy worldwide, and management can be difficult in the locally advanced setting. Here, we examine the literature on the use of neoadjuvant systemic therapy prior to surgical excision of NMSC.

**Abstract:**

Surgery and external-beam radiation therapy are the primary treatment modalities for locally advanced NMSC, but they can lead to impairment of function and disfigurement in sensitive areas such as the head and neck. With the advent of targeted systemic therapies and immunotherapy, physicians have explored the ability to offer neoadjuvant therapy for NMSC in order to reduce surgically induced morbidity. Provided herein is a guide to current applications of neoadjuvant systemic therapies for NMSC and future directions.

## 1. Introduction

Cutaneous malignancies are the most common form of cancer, with a higher incidence in the United States than all other cancers combined [1]. The skin is the largest organ in the body, undergoing constant proliferation while being exposed to ionizing radiation from the sun. This leads to a high rate of neoplastic transformation, as evidenced by estimates of more than 3 million Americans diagnosed with skin cancer annually [2]. Melanoma has a higher mortality rate than non-melanomatous skin cancer (NMSC), including basal cell carcinoma (BCC), cutaneous squamous cell carcinoma (CSCC), and Merkel cell carcinoma (MCC), but is far less common. In an analysis of over 100,000 human cancer genomes, these four cancers sit at the very top of the tumor mutational burden amongst all cancers [3]. While precise statistics are not available for NMSC owing to the lack of a requirement for reporting to cancer registries, there are an estimated nearly 9000 deaths annually due to CSCC alone in the United States compared to the 7650 deaths estimated as a result of melanoma [4]. While mortality from these cancers is relatively rare, the morbidity imparted from disease and treatment can be substantial.

The vast majority of NMSC, with the exception of MCC, can be managed in the ambulatory office setting. The gold standard for curative treatment of NMSC is surgical excision. This can be performed using traditional margin analysis or with the implementation of Mohs micrographic technique. The resectability of locally advanced NMSC, even in cosmetically sensitive locations such as the head and neck, has expanded with the popularization of techniques such as the use of skin substitutes, negative-pressure wound therapy, and microvascular free-tissue transfer. External-beam radiation therapy (RT) may be used in the adjuvant setting for advanced disease or for patients with unresectable disease or with comorbid conditions that preclude surgery. Systemic therapy has traditionally been reserved for palliative treatment of patients with unresectable or metastatic disease. This paradigm has started to shift with the introduction of immunotherapy, which has greater efficacy and reduced toxicity compared to traditional cytotoxic chemotherapy. While no systemic agents are currently approved by the US Food and Drug Administration (FDA) for up-front curative-intent treatment of NMSC, this is an area of active ongoing investigation [5,6,7].

The oldest category of systemic therapies employed for unresectable or metastatic NMSC is chemotherapeutic agents with cytotoxic and anti-metabolic properties. While still used in the setting of treatment-resistant advanced disease, traditional chemotherapy is used less frequently now because of the relative lack of efficacy and high burden of toxicity [8,9,10,11]. Targeted agents have now been developed from the understanding of intracellular signaling pathways that are dysregulated in NMSC. Successful advances against NMSC have been made by targeting epidermal growth factor receptor (EGFR) in CSCC as well as the hedgehog protein signaling pathway in BCC. Other combination systemic agents targeting the signal transduction pathways mediated by tyrosine kinase complexes have been investigated in patients with locally advanced NMSCs [12].

Most recently, immune checkpoint inhibitors have been used in the systemic treatment of NMSC. Checkpoint inhibitors can help the host immune system more effectively combat NMSC by modulating the adaptive immune response and increasing cytotoxic T-cell activity. This class of medication has been proposed to be particularly advantageous in the neoadjuvant setting owing to intact lymphatic circulation and the presence of a higher level of neoantigens compared to adjuvant delivery [13]. Successful targets for checkpoint inhibitors in NMSC include the PD-1/PD-L1 pathway as well as CTLA-4. Given the high mutational burden seen in cutaneous malignancies, it has been proposed that increased immunogenicity associated with mutations may make this class of therapies particularly effective in NMSC [14]. The impressive response to targeted therapy and immune checkpoint inhibitors for unresectable or metastatic NMSC has prompted investigation into their use in the neoadjuvant setting for advanced resectable disease [Table 1].

## 2. Basal Cell Carcinoma

Basal cell carcinoma is the most incident tumor globally and is characterized clinically by a pearly appearance with overlying ulceration and telangiectasia formation. Several variants exist, with infiltrative BCC showing particularly aggressive clinical behavior, making local recurrence and treatment of locally advanced disease an important consideration. This subtype has been shown to be more common in the head and neck [28]. Basosquamous tumors are another clinically important subset of BCC with squamous differentiation. Responses to systemic therapy in basosquamous carcinoma appear to correlate with their cytogenetic profiles, creating the opportunity for neoadjuvant treatment in select cases [29].

### 2.1. Vismodegib and Sonidegib

There are currently two hedgehog inhibitors with FDA approval for use in patients with unresectable BCC. Vismodegib was the first drug of this class, and results of phase I clinical testing were published beginning in 2011 [30]. The phase II ERIVANCE trial included 63 patients with locally advanced disease [15]. Patients were given 150 mg by mouth daily. This cohort had an imaging objective response rate (ORR) of 43%, including a complete response (CR) rate of 21%. A 12-month follow-up to this trial was published in 2015, with a noted increase in ORR from 43% to 48% [16].

Sonidegib, another hedgehog inhibitor, is also an option for unresectable or metastatic BCC. Results of its use in locally advanced disease were demonstrated in the BOLT trial and published in 2015 [17]. This trial included two different dosing protocols, 200 mg and 800 mg by mouth daily. The ORR was 43% among patients receiving 200 mg and 38% among patients receiving 800 mg. The lower dosage group also had half as many (14% vs. 30%) serious adverse events. The most common grade 3–4 adverse events were elevated creatine kinase and lipase levels. A long-term follow-up of this cohort reported an updated ORR of 56% and a median duration of response of 26.1 months [31].

Additional head-to-head data comparing the two available hedgehog inhibitors are needed. They have both been demonstrated to have similar efficacy and tolerability profiles, with the most common adverse events being muscle cramps, alopecia, and dysgeusia [16,17]. There are differences in the pharmacokinetics of the two drugs in regard to the protein binding and serum concentrations, elimination, and volume of distribution, which all influence drug delivery to the skin [32]. The effects of these differences are not completely understood, but both drugs represent promising options in the neo-adjuvant setting. 

### 2.2. Cemiplimab

First approved for the treatment of CSCC, the anti-PD-1 antibody cemiplimab has more recently received FDA approval for patients with locally advanced BCC whose disease has progressed during hedgehog inhibitor treatment or who are not candidates for hedgehog inhibitors. The initial phase II testing of cemiplimab included 84 patients with locally advanced disease, with 31% ORR and 6% imaging CR [18]. Of the remaining patients, 49% had stable disease. Grade 3–4 adverse events occurred in 48% of the patients, with the most common being hypertension, colitis, fatigue, urinary tract infection, and visual impairment. The selected dosing schedule for these patients was 350 mg intravenously (IV) every three weeks. 

### 2.3. Neoadjuvant Therapy

Due to the successful treatment of patients with unresectable or metastatic BCC using hedgehog inhibitors, interest has increased in investigating their use in the neoadjuvant setting in patients with locally advanced, resectable disease. Encouraging evidence for this approach was published in 2021 as part of the VISMONEO study [7]. The intent of this investigation was to explore the use of vismodegib in resectable tumors for which immediate extirpation would result in considerable functional and aesthetic morbidity. Of the 55 patients initially enrolled in the protocol, 4 had inoperable disease, and the remainder were deemed to be curable with surgical ablation with either a major functional risk (*n* = 15), or a major aesthetic or minor functional risk (*n* = 36). Vismodegib was administered daily for 4 to 10 months prior to surgery until a best response was observed. The median treatment duration was 6.0 months. Forty-four (80%) patients had downstaging of their disease with an ORR of 71%. Twenty-seven (49%) patients had an imaging CR. These data support further investigation of a neoadjuvant approach in resectable BCC.

## 3. Squamous Cell Carcinoma

Squamous cell carcinoma is the second most common tumor globally and has a more aggressive clinical behavior than BCC. It is characterized clinically by flesh-colored keratotic papules or nodules that can have an ulcerative component. Given the more aggressive nature of these cancers, wider resection margins are generally employed, which can be relevant in cosmetically and functionally sensitive areas such as the periorbital or perioral regions. 

### 3.1. Cemiplimab 

Cemiplimab was the first FDA-approved systemic treatment for CSCC following the results of the EMPOWER-CSCC-1 trial. Initial data published in 2018 for patients in the metastatic disease cohort were positive, and the outcomes in the locally advanced group published in 2020 confirmed the utility of cemiplimab in this patient population as well [20,33]. In this phase II protocol, 78 patients with CSCC were recruited, who had tumors that were technically unresectable, had experienced two or more instances of recurrence in the same location, had tumors in anatomically challenging locations that would result in severe disfigurement or dysfunction following resection, or had other contraindications to surgery. Patients were treated with weight-based dosing of 3 mg/kg every two weeks. The ORR was 44% with 13% CR. When PD-L1 status was analyzed, 35% of patients with a PD-L1 tumor proportion score (TPS) < 1% and 55% of patients with a TPS ≥ 1 were seen to have an objective response, suggesting that upregulated PD-L1 expression is not necessary for anti-tumor activity.

### 3.2. Pembrolizumab 

The second agent approved for first-line treatment of locally advanced unresectable or metastatic CSCC was pembrolizumab. Pembrolizumab is a PD-1 inhibitor that has been cleared for use in melanoma since 2014, as well as in a wide variety of other tumors [34]. It gained approval for CSCC based on the results of KEYNOTE-629 [22]. Of the one hundred and five patients enrolled, ninety-one (87%) had received prior treatment with systemic therapy. Patients were administered 200 mg of pembrolizumab IV every three weeks for 35 cycles or until progression. Of the 47 patients with locally advanced or recurrent CSCC without distant metastasis, the ORR was 36% with 1 CR. In 2020, Maubec et al. conducted an analysis of the efficacy of pembrolizumab as a first-line agent for unresectable CSCC [23]. This study, known as the CARSKIN trial, initially included 39 patients but was expanded to include an additional 18 and included an evaluation of the impact of PD-L1 expression on response. Patients received 200 mg of pembrolizumab IV every three weeks. The researchers demonstrated an ORR of 41% at week 15 and a CR of 21%. They found that patients with tumors with a TPS < 1% had a significantly lower ORR to therapy (17% vs. 55%). Notably, 32% of patients had disease progression that was noted within the first weeks of treatment.

Given the different inclusion criteria in the respective trials, it is difficult to compare anti-PD-1 therapies [35]. However, it is reasonable to expect that they will perform similarly overall.

### 3.3. Neoadjuvant Therapy

Neoadjuvant therapy for resectable CSCC was first explored using an EGFR inhibitor, gefitinib [19]. More recently, neoadjuvant immunotherapy has been applied to resectable CSCC. In a single-institution pilot study, a 55% pathologic complete response (pCR) and 20% major pathologic response (MPR: > 0% but ≤ 10% viable tumor cells) was observed after two doses of neoadjuvant cemiplimab in 20 patients with resectable stage III-IV CSCC [21]. Based on pathologic response, 12 patients (60%) did not receive adjuvant radiation. At a median follow-up of 42.3 months, none of the patients who achieved a pathologic response had recurred. Importantly, pathologic responders demonstrated improved DFS compared to non-responders (HR 0.092; 95% CI, 0.010–0.886) [36].

The exceptionally high pathologic response rates were recently confirmed in a multicenter, international, phase 2 trial of neoadjuvant cemiplimab in 79 patients with resectable stage II–IV CSCC. During the neoadjuvant period, 62 (78%) patients received four doses of cemiplimab [6]. Two (3%) patients progressed to inoperable disease during neoadjuvant treatment. Response-adapted oncologic surgery was allowed. Based on a centralized independent pathologic review, a pCR was observed in 40 (51%) patients and an MPR in 10 (13%) patients. While long-term oncologic outcomes are not yet available, the safety profile of neoadjuvant cemiplimab was favorable. Grade ≥ 3 adverse events occurred in 14 (18%) patients. There were four fatal adverse events, including one possibly treatment-related death. 

Collectively, these data demonstrate the potential of a neoadjuvant approach to treat advanced-stage, resectable CSCC and highlight the need for a randomized trial [Figure 1].

## 4. Merkel Cell Carcinoma

MCC is considerably less incident compared to BCC and CSCC. The relative rarity of this disease leads to limited research. MCC is characterized clinically by a flesh-colored, well-circumscribed nodule that can demonstrate rapid growth.

There are two distinct types of MCC, one driven by UV-induced mutations and the other by a recently discovered Merkel cell polyomavirus (MPyV) [37]. MPyV is present in up to 80% of MCC. While the high mutational burden serves as the basis of the immunotherapy response in MPyV-negative tumors, virus-specific humoral and cellular immune responses are thought to be the mechanism of response in MPyV-positive tumors [38]. Importantly, both types respond to immunotherapy, and across all anti-PD-(L)1 trials in MCC, response rates appeared similar regardless of tumor viral status [24,39].

### 4.1. Avelumab

The first systemic therapy to be approved for use in MCC was avelumab, a PD-L1 inhibitor. The long-term data for its use in metastatic MCC as a second-line agent after treatment failure with prior chemotherapy were included in the reporting of the JAVELIN Merkel 200 trial by D’Angelo et al. [24]. They administered 10 mg/kg of the drug IV every two weeks until treatment progression. Of the 88 patients enrolled, ten (11%) achieved a CR and an ORR of 33% was seen. An additional 116 patients were recruited for an investigation of the use of avelumab as a first-line therapy and were later reported by the same group [25]. An ORR of 40% was seen in this cohort. Both groups demonstrated a better response rate in those with PD-L1-positive tumors but found that this was not requisite for its use. Although it is not currently approved for patients with locally advanced disease alone, off-label use in this fashion is being investigated [40,41] (Figure 2).

### 4.2. Pembrolizumab

Pembrolizumab was FDA-approved for recurrent locally advanced or metastatic MCC. Evidence for its use in this setting was demonstrated in the Keynote 017 trial, which included 50 patients treated with 2 mg/kg IV every three weeks [26]. Although outcomes were not analyzed according to the presence of metastatic disease versus advanced locoregional disease, the ORR was 56%, including 24% demonstrating an imaging CR. Within this study, the authors found that tumor response was not correlated with PD-L1 expression.

### 4.3. Retifanlimab

Retifanlimab is a monoclonal antibody to the PD-1 receptor. The POD1UM trial investigated its use in patients with metastatic or locally advanced MCC [27]. Eighty-seven treatment naïve patients were enrolled and received 500 mg of retifanlimab IV every four weeks for up to two years. Per protocol analyses of the initial 65 patients enrolled demonstrated an ORR of 46% and a CR in 12%. These data led to FDA approval in this setting. Final results from this study are pending, but the initial data have stimulated interest in this medication as an additional option for those with locally advanced disease.

### 4.4. Neoadjuvant Therapy

CheckMate 358 was a trial using nivolumab, an anti-PD-1 agent, for the treatment of various cancers, including resectable MCC [5]. The researchers investigated a two-dose course of 240 mg of nivolumab IV with planned surgical excision on day 29 of treatment. Of the 39 MCC patients enrolled, all but three underwent eventual surgical excision. Overall, seventeen (44%) had a pCR. A central review yielded an additional four (11%) patients with MPR, which was not evaluated in the initial site review. One patient had disease progression during treatment, and two other patients who did not complete the protocol withdrew because of grade 2 non-treatment-related nausea and grade 3 treatment-related rash. Additional studies are looking at neoadjuvant treatment of resectable MCC with pembrolizumab plus lenvatinib and cemiplimab alone (NCT04869137, NCT04975152).

**Figure 2 cancers-15-05494-f002:**
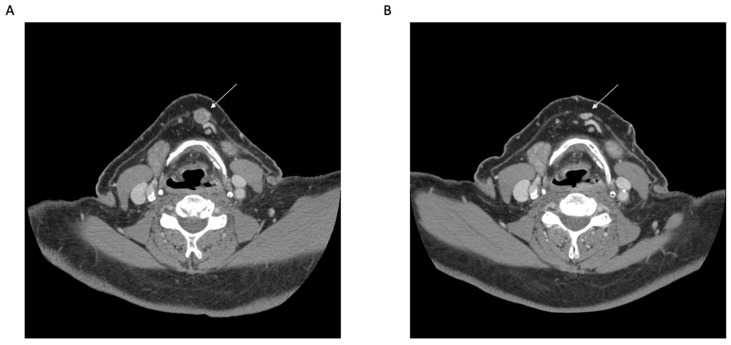
Neoadjuvant immunotherapy for node-positive Merkel cell carcinoma. Representative axial computed tomography (CT) images before (**A**) and after (**B**) neoadjuvant immunotherapy (Avelumab) for recurrent Merkel cell carcinoma metastatic to a left submandibular lymph node (arrow). Histologic interpretation after surgery demonstrated a complete pathologic response.

## 5. Other Considerations

### 5.1. General Considerations

Multidisciplinary management of complex NMSC patients is essential. Successful long-term, high-quality outcomes are a result of proper systemic therapy, integration of timed surgical intervention, and judicious use of radiation therapy. The details of these considerations follow below. 

### 5.2. Patient Selection

Patients with unresectable and/or metastatic NMSC are currently eligible to receive systemic therapy. Patients with ‘borderline’ resectable NMSC may be considered for neoadjuvant systemic therapy with the goal of shrinking the cancer to allow for function-preserving, curative-intent surgery. This approach is most warranted in cases where the functional morbidity of treatment is extreme (e.g., orbital exenteration). Discussion should be undertaken with patients to understand the goals of treatment, risks, and any other factors that might influence their care. Management should be discussed in a multidisciplinary setting with input from medical, surgical, and radiation oncology. The overall fitness of a patient and the goals of therapy should be carefully considered for each individual. 

Immunosuppression is generally considered a contraindication to neoadjuvant immunotherapy. Given the high rates of development and poor outcomes of NMSCs in patients with a history of solid organ transplant requiring iatrogenic immunosuppression, this patient population is of particular interest. While some data are available regarding the use of checkpoint inhibitors in these patients, further investigation is needed before routine use can be implemented [42]. 

In prior trials, response to various systemic agents based on alterations in cell signaling pathways and protein expression has been investigated. Response rates to checkpoint inhibition were generally more favorable in individuals with increased PD-1 or PD-L1 expression, but upregulation was not a requisite for success and was not seen to correlate with response in MCC. Thus, PD-1 or PD-L1 expression alone is insufficient for selecting patients for treatment with immunotherapy, including neoadjuvant approaches. 

### 5.3. Surgical Timing

Currently, the optimal timing of surgery after neoadjuvant therapy is not established and is dependent on several factors, including the histology of the disease, resectability at presentation, selected systemic agent(s), and durability of individual treatment response. There are two main neoadjuvant treatment strategies to consider. The first is a pre-determined surgical deadline at which systemic therapy is stopped and the tumor is excised. This approach is supported by prospective clinical trials but varies based on histology. In resectable CSCC, for example, there appeared to be no significant difference in response rates between two and four cycles of neoadjuvant cemiplimab. In contrast, resectable BCC patients were typically treated for months in VISMONEO. Patients with adverse events precluding the continuation of systemic therapy can require surgery before the planned interval. The second strategy of surgical timing is that of continued administration of systemic therapy until tumor regression ceases or a complete clinical and radiographic response is achieved. The goal of this approach is to decrease the morbidity of eventual surgical intervention as much as possible, but it increases the risk of progression and toxicity from treatment. 

### 5.4. Extent of Surgery

It is not known exactly how much clinical and radiographic responses to neoadjuvant therapy can reduce the ultimate volume of tissue resected. It has been demonstrated that in breast carcinoma, magnetic resonance imaging (MRI) can serve as a helpful guide for assessing the extent of the tumor after neoadjuvant chemotherapy [43]. NMSCs are not easily assessed with cross-sectional techniques, and imaging responses can underestimate pathologic responses. For example, in resectable CSCC, an imaging CR was observed after neoadjuvant immunotherapy in only 5 (6%) patients, while 40 (51%) demonstrated a pCR [6]. 

Response-adapted oncologic surgery can be considered in NMSC patients who respond to neoadjuvant therapy. The principles of response-adapted oncologic surgery include the following:Surgical plan that may be adapted to tumor response after neoadjuvant treatment;Planned R0 resection around gross residual visible tumor and any imaging abnormality after neoadjuvant treatment;Intraoperative margin control of primary tumor (recommended from main specimen rather than surgical defect);Resection of all first echelon draining nodal basins and levels based on baseline imaging;Preservation of functional structures acceptable if confirmed as histologically negative in adjacent tissue.

## 6. Conclusions

NMSC is common and can lead to extensive morbidity from local therapies, especially in the head and neck. Traditional treatments for locally advanced disease, namely surgical excision, can impart significant impairment in form and function. The use of neoadjuvant systemic agents may allow for the surgical treatment of previously unresectable disease and the possibility of improved quality of life in individuals with locally advanced disease. The greatest responses have been observed with the use of neoadjuvant immunotherapy, which provides the framework to improving long-term outcomes in patients with NMSC. Future research will better define the role of neoadjuvant therapy for NMSC. 

## Figures and Tables

**Figure 1 cancers-15-05494-f001:**
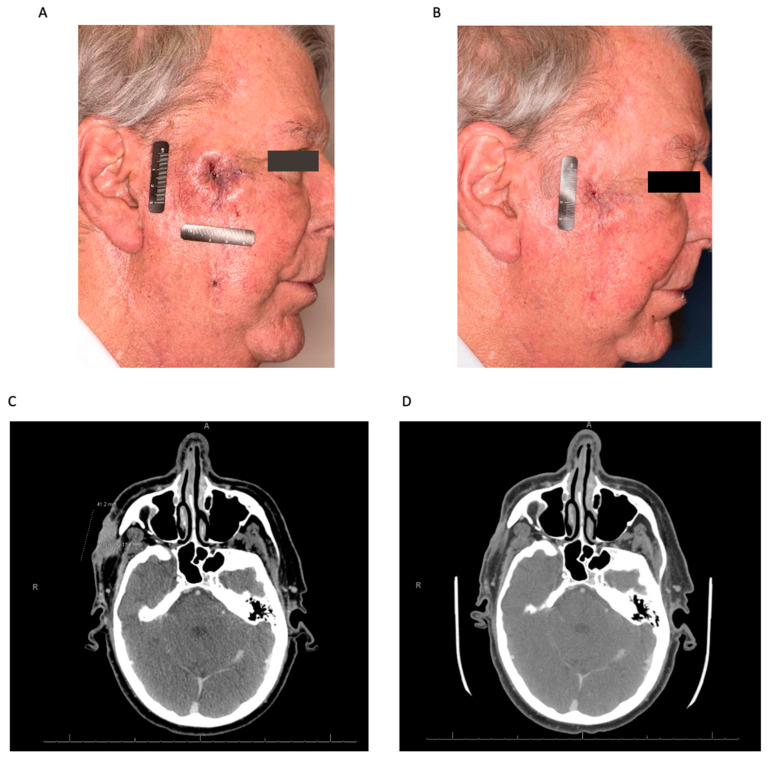
Neoadjuvant Immunotherapy for Cutaneous Squamous Cell Carcinoma. Representative clinical photographs (**A**,**B**) and axial computed tomography (CT) images (**C**,**D**) before (**A**,**C**) and after (**B**,**D**) neoadjuvant immunotherapy (Cemiplimab) for cutaneous squamous cell carcinoma involving the right cheek. Histologic interpretation after surgery demonstrated a complete pathologic response.

**Table 1 cancers-15-05494-t001:** Responses to checkpoint inhibitors and targeted therapy in NMSC.

		Locally Advanced/Unresectable			Neoadjuvant		
	Agent	ORR	CR		ORR	MPR + pCR	pCR
BCC	Vismodigib	43–48% [15,16]	21% [15]		71% [7]	NR	NR
	Sonidegib	38–56% [17]					
	Cemiplimab	31% [18]	6% [18]	2nd line			
CSCC	Gefitinib				45% [19]	NR	NR
	Cemiplimab	44% [20]	13% [20]		30–68% [6,21]	64–75% [6,21]	51–55% [6,21]
	Pembrolizumab	36–41% [22,23]	2–21% [22,23]				
MCC	Avelumab	33–40% [24,25]	11% [24]				
	Pembrolizumab	56% [26]	24% [26]				
	Retifanlimab	46% [27]	12% [27]				
	Nivolumab				46% [5]	NR	44% [5]

ORR, objective response rate; CR, complete response; MPR, major pathologic response; pCR, pathologic complete response; BCC, basal cell carcinoma; NR, not reported; CSCC, cutaneous squamous cell carcinoma; MCC, Merkel cell carcinoma.

## Data Availability

The data presented in this study are available in this article.

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
