# Peer review of "Neoadjuvant Approaches to Non-Melanoma Skin Cancer"

_cancers, 2023, doi:10.3390/cancers15235494_

Round 1

Reviewer 1 Report

Comments and Suggestions for Authors

The article is clear and well written

About tables and figures, there is only one figure, corresponding to a coherence tomography of a non-resectable Merkel cell carcinoma before and after systemic treatment with Avelumab (PD-L1 inhibitor) as neoadjuvant. It is clear, except that in the caption they could indicate that it corresponds to treatment with Avelumab (the agent is not indicated).

The Table is essential as a summary of data from the included clinical trials. As for the figure, it is representative. However, I would include another image of another type of NMSC before/after neoadjuvant treatment.

Reviewer 2 Report

Comments and Suggestions for Authors

The Authors present a report about neoadjuvant approaches to non-melanoma skin cancers. Although the manuscript is of interest, some minor changes area needed:

- In the Introduction when you present Table 1, you should specify is these are results from the literature and in case the relative citations are needed.

- In their relative paragraphs,  should add some sentences (maybe 5 rows) about the clinical features of basal cell carcinoma, squamous cell carcinoma, Merkel cell carcinoma. Maybe you can describe (briefly) what are the main clinical  features of these malignancies.

- Please, always in the Basal cell carcinoma's paragraph, please add some sentence about the fact that thet are some basal cell carcinomas, that (regardless the size and local invasiveness) may show aggressive clinical and histologic features and for this reasons this type of  basal cell carcinoma they must be followed more closely and they can benefit from the use of neoadjuvant therapy, especially when localized in certain anatomical places, they must be followed more closely and they can benefit from the use of neoadjuvant therapy, especially when localized in certain anatomical places. In this regard, please read and add to your literature this recent article "Clinical and Dermoscopic Factors for the Identification of Aggressive Histologic Subtypes of Basal Cell Carcinoma. Front Oncol. 2021 Feb 19;10:630458PMID: 33680953; PMCID: PMC7933517."

- Page 3 line 10: there is a typo, please change Sinodegib into Sonidegib

- Please highlights the main differences between Vismodegib and Sonidegib, also regarding the effectiveness and relative side effects.

- Please add some sentences about also "Basosquamous carcinoma" and relative managment with neoadjuvant therapy.

Reviewer 3 Report

Comments and Suggestions for Authors

In their paper ‘Neoadjuvant approaches to non-melanoma skin cancer’, the authors attempt to provide a comprehensive overview of the literature regarding the use of neoadjuvant systemic therapy in NMSC.

In general, Zelin et al have already published a similar review regarding this topic (Curr Treat Options Oncol. 2021; 22(4): 35.). The authors need to be more distinctive to this work in their review. I have serious issues regarding the completeness of this review. Many treatment options have not adequately been addressed.

Simple summary

Please rephrase the first sentence (change in to something like ..has highest incidence..).

Rephrase ‘surgical ablation’ into surgery

Introduction

Line 60, please refrain from naming drugs, rather name the signaling pathway they target

Table 1. This table is not complete. The authors did not include the (modest) responses to Erlotinib (Cancer. 2018;124(10):2169.) in CSCC, nor did they include the responses to Dacomitinib, Cetuximab and Lapatinib. Response rates to classic chemotherapy or even the old itraconazole are also lacking.

What is the point of including a table that does not cover the entire spectrum of drugs that have been tested in a neoadjuvant and/or locally advanced metastatic setting?

Furthermore, it is Sonidegib (not Sinodegib)

Basal Cell Carcinoma

Why did the authors not compare both Hedgehog inhibitors? Although trials that directly compare the two drugs do not exist to my knowledge, there are differences in pharmacokinetic profile potentially affecting response rate and side-effects (J Eur Acad Dermatol Venereol. 2020;34(9):1944-56.)

The authors do not discuss the use of other checkpoint inhibitors besides cemiplimab in BCC (J Immunother Cancer. 2022; 10(12): e005082). This is an omission. Although trials on neoajuvant treatment are lacking, one could assume based on the available data that response to other neoadjuvant checkpoint inhibitors is likely.

Squamous Cell Carcinoma

Data on (neoadjuvant) nivolumab (mostly case series) are lacking in this paper. In order to be a comprehensive review, these data should be addressed (see for overview Curr Treat Options Oncol. 2021; 22(4): 35; )

Merkel Cell Carcinoma

The authors have not addressed the use of chemotherapy in MCC.

Other Considerations

If the use of neoadjuvant systemic treatment will increase in the near future, some comments regarding toxicity of the treatment should be made. Furthermore, and I think even more important, the topic ‘patient selection’ needs to be elaborated, especially the problems one encounters with the specific population of people with NMSC. This population is generally old, and very often has immunosuppressants (patients after solid-organ transplants, patients with HIV etc). Many of these factors seriously complicate neoadjuvant immunotherapy.

I do miss Future Directions. Where do we go from here?

Overall, this paper has serious flaws and I do not think that it merits publication in Cancers.

Comments on the Quality of English Language

Minor editing of English language required

Reviewer 4 Report

Comments and Suggestions for Authors

The authors present an essential overview of treatment options with focus on neoadjuvant setting in non-melanoma skin cancer. I have no major issues, I would only suggest: 1. in the Table, please report the references from where the data in the table are derived, because as it is it is not clear if the authors reported ranges, average values o exact data 2. in the part of patient selection, discuss a bit more the differences observed in clinical trials between patients with cancer expressing and not expressing PD-L1, similarly to what present in general HNSCC eg. are there proposed cut-offs of pD-L1 expression for immunotherapy administration? If not why? I feel that this point is worth a little more discussion.

Round 2

Reviewer 3 Report

Comments and Suggestions for Authors

The authors have revised the manuscript substatially, however I still have major issues that need to be addressed before the manuscript is ready for publication.

The title is misleading and should be altered. This review is not a comprehensive review of the literature on neoadjuvant therapy of NMSC it only covers targeted therapy and immunotherapy and only those drugs that are FDA approved. This should be made very clear throughout the manuscript. Rather than using the general term 'systemic therapy' the authors ought to use 'immune checkpoint inhibitors' and/ore 'targeted therapy' (correctly) where appropriate.

Since the authors focus on immune checkpoint inhibition (aside targeted therapy) as a neoadjuvant approach they should elaborate why this is a logical approach. Indeed they cover surgical irresectability, but a more in depth discussion on the rationale of neoadjuvant immunotherapy would improve the manuscript (either in the Introduction or Discussion). The antitumour activity of immune checkpoint inhibitors requires the expansion and broadening of the tumour-specific T cell population. Therefore, administering immune checkpoint inhibitors prior to surgery enhances their efficacy (for more info regardig this topic in melanoma see Nature Reviews Clinical Oncology 2023; 408-422). 

volume20pages

In line 72 'targeted therapy and' should be added before 'immune checkpoint inhibitors'. The title of table 1 must also be altered to 'Responses to targeted therapy and immune checkpoint inhibitors in NMSC'.

The authors have only once corrected Sinodegib to Sonidegib (line 98), while throughout the manuscript the repeat the same error (see for instance table 1 and line 89).

In line 105 the authors suggest that vismodegib and sonidegib are checkpoint inhibitors, they are not, please correct this flaw.

Round 3

Reviewer 3 Report

Comments and Suggestions for Authors

The manuscript had been altered sufficiently.